# Imaging the Infection Cycle of T7 at the Single Virion Level

**DOI:** 10.3390/ijms231911252

**Published:** 2022-09-24

**Authors:** Bálint Kiss, Luca Annamária Kiss, Zsombor Dávid Lohinai, Dorottya Mudra, Hedvig Tordai, Levente Herenyi, Gabriella Csík, Miklós Kellermayer

**Affiliations:** 1ELKH-SE Biophysical Virology Research Group, Tűzoltó Str. 37-47, H1094 Budapest, Hungary; 2Department of Biophysics and Radiation Biology, Semmelweis University, Tűzoltó Str. 37-47, H1094 Budapest, Hungary

**Keywords:** single-particle imaging, TIRF, total internal reflection fluorescence (TIRF), AFM, atomic force microscopy (AFM), virus docking, 2D walk, bacterial lysis

## Abstract

T7 phages are *E. coli*-infecting viruses that find and invade their target with high specificity and efficiency. The exact molecular mechanisms of the T7 infection cycle are yet unclear. As the infection involves mechanical events, single-particle methods are to be employed to alleviate the problems of ensemble averaging. Here we used TIRF microscopy to uncover the spatial dynamics of the target recognition and binding by individual T7 phage particles. In the initial phase, T7 virions bound reversibly to the bacterial membrane via two-dimensional diffusive exploration. Stable bacteriophage anchoring was achieved by tail-fiber complex to receptor binding which could be observed in detail by atomic force microscopy (AFM) under aqueous buffer conditions. The six anchored fibers of a given T7 phage-displayed isotropic spatial orientation. The viral infection led to the onset of an irreversible structural program in the host which occurred in three distinct steps. First, bacterial cell surface roughness, as monitored by AFM, increased progressively. Second, membrane blebs formed on the minute time scale (average ~5 min) as observed by phase-contrast microscopy. Finally, the host cell was lysed in a violent and explosive process that was followed by the quick release and dispersion of the phage progeny. DNA ejection from T7 could be evoked in vitro by photothermal excitation, which revealed that genome release is mechanically controlled to prevent premature delivery of host-lysis genes. The single-particle approach employed here thus provided an unprecedented insight into the details of the complete viral cycle.

## 1. Introduction

Although in recent years viruses infecting humans have gained increased interest, plant viruses also present a major problem in food and agricultural industries, which indirectly influences our everyday life [1,2]. Understanding the host-recognition mechanisms and the molecular steps of the infection cycle of viruses is fundamental in the development of proper anti-viral strategies, as well as utilizing bacteriophages as alternatives to antibiotic treatments [3]. Bacteriophages are viruses that attack bacteria. Among them, T7 bacteriophages are dsDNA viruses comprised of an icosahedral capsid with an internal protein core and a short non-contractile tail. This tail-fiber complex enables the phage to bind to its target and deliver its genome to the host [4]. Although the structure and properties of T7 have been investigated extensively [5,6,7], our knowledge of its host recognition and DNA injection mechanisms remains incomplete [8,9]. A recent cryo-EM study suggested that the tail-fiber complex might play a crucial role in docking and anchoring the bacteriophage to its target cell [10]. The fibers were proposed to reversibly bind to either the capsid shell or the target cell surface and aid the search in finding the final target receptor. The search is thought to be terminated by the binding of the tail to its receptor which then prompts the phage to eject its DNA. DNA is translocated across the bacterial envelope into the target cell through an elongated channel assembled from the viral core proteins. Although the cryo-EM images provided details with unprecedented resolution, the dynamics of infection could be only indirectly inferred.

The driving forces and mechanisms utilized in the initial part of DNA injection have long been debated. Ejection is thought to be driven either by the extreme pressure of DNA packed compactly inside the capsid or by an active mechanism that would translocate the initial part of the genome through the host’s membranes [9,11]. Further DNA translocation is first carried out by host proteins, then viral enzymes such as RNA polymerases are thought to take over [8]. In vitro DNA ejection studies have shown that viral genome release can be a strictly controlled, multistep process [12,13]. In a unique approach, the DNA injection process of individual lambda phages was observed by fluorescently labeling the viral DNA [14]. This study revealed significant cell-to-cell variability in the kinetics of DNA translocation which would have remained hidden in bulk experiments. The experiment pointed out that the presence and status of the target cell cause high variability in the speed hence the timing of DNA translocation (1–20 min to completion), compared with in vitro DNA ejection which was more uniform. In the case of the lambda phage, the genomic delivery might not be completely enzyme driven since the rate of DNA translocation decreases throughout the injection, dropping from ~20 kbp/min to zero. It has also been pointed out that the rate of DNA translocation in T7 appears to be more constant hence likely driven by enzymes [8].

Following successful genome translocation, viral replication is initiated, and new phages are formed. In the case of dsDNA bacteriophages, packaging the genome replicas into freshly formed viral shells is a closely controlled and intricate process [15]. The final stage of infection involves the release of the freshly assembled viral progeny. In the case of T7, lysis is a multistep process: initially, holins start to form pores in the inner membrane allowing the passage of peptidoglycan-digesting enzymes, finally, a spanin-mediated, co-localized fusion of the inner and outer membranes leads to violent, burst-like lysis [16,17]. A recent study has shown that the bacterial envelope undergoes significant changes prior to lysis [18]. Following viral infection, membrane blebbing and vesicle formation takes place, the role of which is yet to be determined.

Here we investigated three important steps of the infection cycle of T7: bacteriophage docking, DNA translocation, and virally induced bacterial lysis. Although bulk experiments have uncovered significant detail about these steps [11,19], due to ensemble averaging the exact mechanisms remained hidden [20]. To circumvent the problems of ensemble averaging, here we used single-particle approaches to localize individual bacteriophages on target-cell surfaces, follow their in-situ motion, explore their DNA translocation process, and follow the structural dynamics of the host-cell lysis.

## 2. Results

In this study, we investigated three crucial steps of the T7 phage infection cycle by using single-particle tools: search and binding-site recognition of the virus on *E. coli* surface, DNA delivery into the host, and lysis of the host cell.

To optimize the environmental circumstances of the single-particle assays, we first quantitated the effects of culture media, phage-to-cell ratio (i.e., multiplicity of infection, MOI), and DNA-staining dye concentration on the growth rate of bacterial populations. If we used LB medium, which is optimal for bacterial growth, then lysis occurred after ~30 min in a synchronized manner. By contrast, in PBS, which is more suitable for TIRF and AFM measurements, the growth rate declined strongly, and synchronized lysis could not be observed within 120 min (Figure 1A). Even though bacteria were unable to multiply in PBS they remained alive and accessible for infection. The phage-to-cell ratio also affects lysis dynamics (Figure 1B) [21] due to the fine balance between the number of accessible resources set by host metabolism and the demand set by the number of phages simultaneously infecting the cell. At low MOI (~1) bacterial growth proceeded slowly and was not interrupted (Figure 1B gray trace). At very high MOI (~500) the bacterial population declined progressively due to over-infection, and deterministic lysis needed for single-particle analysis was absent (Figure 1B blue trace). At an intermediate MOI of 50 a synchronized, deterministic lysis could be observed after ~40 min (Figure 1B red trace). Because the estimated ratio of infective to non-infective phages was as low as 1:5 (determined by viral titration), the true MOI was estimated to be about 10. In subsequent experiments, this adjusted MOI was used. Finally, we tested the effect of SyO on infection dynamics and observed that lysis was only slightly delayed and prolonged (Figure 1C).

### 2.1. T7 Explores the E. coli Surface for Binding Site 

To follow the target recognition kinetics and DNA translocation process of bacteriophages, their DNA was labeled with SyO by allowing the dye to penetrate the capsid shell. Docked bacteriophages appeared as bright fluorescent particles along the edge of the bacterium (Figure 2A). Phage binding to *E. coli* was isotropic, as we could not observe any directional or spatial preference (Appendix A). To reveal the surface dynamics of individual phage particles, we tracked the position of the fluorescent spots as a function of time. In the early stages of infection, bacteriophages that retained their genomes long enough to be observed for several minutes moved around randomly on the target cell surface (Appendix A, Figure 2A,B). Although phage movement occasionally accelerated (Figure 2A/2,C) and eventually stopped altogether (Figure 2B), initial periods of random motion with a more-or-less constant velocity were well discernible. The displacement of the phage particles from their landing point as a function of time (Figure 2C) could be fitted with a power function with a fractional exponent, suggesting diffusive motion. At room temperature, the random target search proceeded for several minutes (Figure 2C,D). The total travel path grew linearly and could reach up to 5 µm (Figure 2D). The mean slope of this function was ~10 nm/s. The step size of the motion, measured as the displacement of particle position between consecutive video frames spaced 1 s apart, was normally distributed with a mean of ~10 nm (Figure 2E), supporting the finding that T7 particles search for their target site with an average speed of 10 nm/s. To test whether the target search is free diffusion, we measured, with identical analytical tools, the two-dimensional diffusion of polystyrene beads with radii similar to that of T7 (Figure 2E). Although the overall shape of the bead step size distribution was similar to that of the T7 particles, notably, the mean was shifted to 5 µm, indicating that the average speed of bead motion was nearly two orders of magnitude greater than that of target search by T7. Thus, target search is a diffusive motion, the speed of which is limited, most likely, by reversible interactions between the virus and the host cell surface.

### 2.2. Fate of Docked T7 and Its Genome

To reveal the fate of individual docked phage particles and their genomic DNA, we constructed circumferential kymograms by plotting the intensity profile along the bacterium’s edge as a function of time (Figure 3A,B). The fluorescence intensity of the T7 particles decreased as a function of time. Depending on the kinetics of the intensity drop we identified three different processes taken by the virus particles or their DNA (n = 36). First, and most frequently (53% of events), intensity decayed with an exponential process, which corresponds to the injection of viral DNA into the target cell (Figure 3A–C, e.g., phage a). Second, intensity sometimes dropped but then increased and fluctuated (25%). This process corresponds to “misfiring” by the T7 phage during which the genomic DNA is ejected into the surroundings rather than into the host cell (Appendix A, Figure 3A,B, phage b). Interestingly, misfiring became more frequent in the later stage of infection, and floating, partially ejected DNA strands could be seen around docked bacteriophages (Appendix A). Third, the fluorescence signal sometimes disappeared within a single frame, which corresponds to unbinding of the T7 phage particle from the host surface (22%), whereas in the case of viral exploration the signal moved along the bacterial periphery The finding indicates that partial target binding of T7 is reversible. We note that rapid misfiring events (*c.f.* Figure 3A(b,b*) phages) were sometimes difficult to distinguish from quick unbinding events.

### 2.3. DNA Ejection Can Be Induced by Photothermal Activation

To investigate the DNA ejection by T7 bacteriophages in vitro, we investigated single phage particles immobilized in a microfluidic device and monitored the appearance of DNA by SyO labeling and TIRF microscopy. In the beginning of the experiment the field of view was dark, and bright spots corresponding to single T7 particles started to appear with a lag time due to in situ labeling with SyO (Figure 4A, Appendix A). After an average lag time of 21 s (±12 s; n = 34) viral DNA was ejected into the medium, where it was immediately stretched out in parallel with the flow direction. DNA ejection was a rapid process, and more than half of the genome was ejected within a fraction of a second (0.13 ± 0.07 s) (Appendix A).

Ejection was never complete, and the majority of the viral genome (84 ± 5%; n = 134) remained attached to the capsid throughout the observation period (Figure 4B). 82% of DNA ejections halted after 80% of the genome has been released. 

### 2.4. Structural Dynamics of the Infected Host Cell Surface 

TIRF microscopy allowed the visualization of the docking process of bacteriophages with a high temporal resolution, but it was only able to image the coverslip-facing side of immobilized bacteria, and only to a depth of ~200 nm. Considering that the circumferential diameter of an *E. coli* cell is about 800 nm (measured with AFM) cell surface events remain hidden. To reveal the topographical changes of the host cell caused by T7 infection, we scanned their surface with high-resolution AFM (Figure 5). AFM, similarly, to TIRF microscopy, revealed an increasing number of docked phages, which were identified by their consistent 60 nm height (Figure 5A). Viruses saturated the bacterial surface with time (~1 h at room temperature) (Appendix A). Often, phages detected in an AFM image were no longer present at the same location in the subsequent AFM scan.

To achieve more stable imaging and improved cell membrane stability, chemical fixatives were added to stabilize bacterium-to-surface as well as phage-to-bacterium adhesion. It must be noted, however, that the process of chemical fixation involves multiple washing steps which might accidentally lead to the dissociation of weakly adhered bacteriophages. Snapshots revealed a surface covered by bacteria at different stages of infection, depending on the moment of fixation. Samples fixed after 15 min consisted of intact bacteria which had small bumps extending from their membrane surface (Rq = 5.2) (Figure 5B). These extending bumps were identified by their height (~60 nm protrusions) and shape to be individual T7 bacteriophages (Figure 5B, inset). Bacteriophages showed no site preference in docking, as they were distributed evenly on the bacterial surface. If the infection was allowed to proceed for 30 min (Figure 5C), the number of successfully docked phages per bacteria increased to an average of 10 phages per bacterium, which at this point matched our calculated starting ratio (MOI = 10) (Figure 6). After 30 min the bacterial surface became rougher (Rq = 11.4), which was never observed in non-infected bacteria (Appendix A). After 45 min, the lysis of individual bacteria started in a highly synchronized manner. The final stages of infection were captured on samples which were fixed after 60 min (Figure 5D). At this point, a large portion of the surface-bound bacteria had already been lysed, and only their remnants were found. Many of the bacteria which were still intact, appeared morphologically very different from those seen at earlier stages. Their surface roughness increased further (Rq = 18.2), and membrane protrusions (blebs) started to appear. These blebs had a diameter of 246 ± 62 nm. In some cases, bacteriophage-to-host connections were stable enough to allow high-resolution scanning, which revealed the nanoscale details of viral infection (Figure 5D/1). In the 3D representation of these images, individual bacteriophages could be identified by their tail fibers attached to the host cell in a similar configuration seen in cryoEM images (Figure 5D/2) [10]. The number of docked bacteriophages started to decrease after 45 min to an average of six phages per bacterium, compared to the steady increase seen in TIRF images (Figure 6). The apparent decline in phage number is most likely directly related to bleb formation that prevents the topographical detection of phage particles but allows for detection by fluorescence.

### 2.5. T7 Infection Induces E. coli Blebbing and Leads to a Violent Lysis

To observe the dynamics of bleb formation, we followed infected *E. coli* cells by using phase-contrast microscopy (Appendix A). We were able to follow the formation of individual bacterial bleb formation with a time resolution exceeding that of AFM (Figure 7A,B, Appendix A). However, due to spatial resolution limitations, only large blebs could be observed. The phase-contrast recordings showed the formation of expanding bacterial blebs at various sites of the bacterial surface on a timescale of a few minutes. Blebs were an order of magnitude larger than those identified by AFM. If SyO, a membrane impermeable nucleic acid dye used for dead cell staining, is added to the sample, then lysis is immediately followed by a bright fluorescent explosion (Figure 7C).

## 3. Discussion

### 3.1. T7 Explores the E. coli Surface for Binding Site 

We suggest two possible mechanisms to explain T7′s diffusive target search. First, viral binding to the host cell is a dynamic process during which the fibers repeatedly bind to and unbind from the bacterial surface, thereby allowing the phage to search for a final binding site. The process thus resembles two-dimensional *walking*, whereby the number of fibers interacting with the surface progressively grows until a stable surface attachment is achieved. Second, and more likely, bacteriophages carry out an initial *rolling* on the bacterial surface until the tail binds to its receptor. Then the fibers are released from their original, capsid-attached arrangement and bind to the bacterial surface to stabilize the T7 capsid in a position optimal for genome ejection and delivery. We note that while both mechanisms require a dynamic surface adhesion during target search, they predict different time-evolution of speed: during *walking* the speed progressively drops; by contrast, during *rolling* it stays constant. A two-dimensional walking mechanism has been proposed before for T7 on theoretical grounds but with no direct experimental evidence [10]. In this model, the fibers play a key role in the initial anchoring of T7. However, if the number of surface-bound fibers increased with time, then the exploration process should be progressively decelerating, which is in contrast to our experimental finding (Figure 2D). In our alternative model, T7 phages roll on the surface at an essentially constant speed until the tail binds to its receptor. Once the tail is bound, the fibers are released from the side of the capsid and assist in the final positioning and anchoring of the phage to the host. The predicted constant speed is well supported by our experimental results (Figure 2D). In this mechanism the fibers play an important role not so much in the target search but in proper capsid positioning, so that genome transfer can be faithfully executed. Indeed, fibers are essential for successful infection, as fibreless mutants are non-infective [22]. Our model thus presents a novel sequence of binding actions in the viral target search mechanism, proposing that fibers are secondary receptors responsible for stabilization and precise capsid orientation.

### 3.2. Fate of Docked T7 and Its Genome

The viral delivery of fluorescently labeled DNA into the target cell has been investigated before and shown to result in an increased overall intracellular fluorescence intensity [14]. By contrast, we could not measure a noticeable intensity increase inside the infected cells [14]. The lack of intracellular fluorescence intensity increment may be explained by the dispersion of the injected DNA in the host cell, hence its escape from the shallow excitation field (roughly 7% of the total volume of an 800-nm-high bacterium is imaged in TIRF). Alternatively, the intercalated DNA might not pass through the narrow channel formed by the elongated tail of the bacteriophage, and the dye molecules are forced to dissociate. The disappearance of fluorescence can also be described by the presence of DNA-dependent polymerases inside the host cell, which would knock out any intercalators while unwinding the DNA double helix. In comparison to lambda phage DNA translocation (1–20 min), T7 was much quicker and more efficient (90% drop in integrated fluorescence intensity in 1 min). Such accelerated DNA translocation might be explained by a higher pressure inside the T7 capsid (60 atmospheres) compared to lambda (15–20 atmospheres) [23,24]. A major difference between these two types of phages is that lambda has a long tail without any internal proteins, whereas T7 has a short tail and internal core proteins, which might play a crucial role in further accelerating DNA translocation across the bacterial membrane [10]. At later stages of infection (30–40 min), phages were still binding to target bacteria; however, no decrease in fluorescence intensity was apparent, indicating that DNA translocation became inhibited by then. Since at this stage all bacteria are expected to be successfully infected by at least one phage, their internal enzymatic machinery has already been drastically reprogrammed and further DNA translocation might be completely inhibited.

### 3.3. DNA Ejection Can Be Induced by Photothermal Activation

We hypothesize that the mechanism of phage activation is related to the local heating of the capsid through the absorbance of incorporated DNA-intercalating dye which penetrated the capsid. The observed DNA ejection corresponds to “misfiring” since it started without the presence of the viral receptor. DNA ejection abruptly stops after 80% of the genome is ejected. We suggest that this halting mechanism has relevance in the natural infection process of the T7 phage. Lysis-related class III genes are transcribed starting from 91% of the viral genome (location of first lysis gene, type II holin: 36344/39936 bp, corresponds to 91%) [25]. Presumably, the terminal region of the genome carries innate signals that hinder the release of the terminal segment, thereby delaying the transcription of lysis-related genes. How this mechanical barrier is overcome is yet unclear. Regardless of the molecular mechanism, our finding reveals the presence of a mechanical control that tunes the timing of the host cell lysis hence the infection cycle.

### 3.4. Structural Dynamics of the Infected Host Cell Surface

Phages found on the surface of a host cell often disappeared in the next AFM scan (Figure 5A). This finding substantiates the notion that T7 phage particles carry out a surface exploration that results in changing positions. Alternatively, the AFM scanning may have mechanically knocked off some of the T7 particles that were weakly bound. This explanation predicts that AFM scanning functions as a mechanical disinfectant that slows down the infection cycle. In support of this idea, we observed that most of the bacteria outside the scanning area lysed during the measurements, yet the bacterium that was being scanned did not lyse during a one-hour-long experiment. 

Bacteria displayed surface protrusions as a result of infection by T7 (Figure 5D). Bleb formation can only be attributed to viral infection since it never appeared on the surface of non-infected bacteria. Similar bleb formation has been seen as a response to antibiotics, and it has also been observed as a bacterial response to infection by T7 [18,26].

### 3.5. T7 Infection Induces E. coli Blebbing and Leads to a Violent Lysis

The size difference in bleb sizes observed by AFM and phase-contrast microscopy has a possible explanation: AFM images might only reveal blebs at the start of their formation. The appearance of the blebs is not unique to infection by T7 but can appear as a result of different external factors. One of these effects is the addition of antimicrobial peptides, which can initiate hole formation in the cell wall or the outer membrane [26,27]. The morphology of blebs induced by the addition of antibiotics is similar to the ones appearing as a result of viral infection. These antibiotics can lyse bacteria by forming a pore in the cell wall, which leads to membrane bulging and thereby lysis of the cell [27]. Infection by T7 leads to a similar outcome, but the course of action is somewhat different and bleb formation is not mandatory for lysis (Figure 7C, Appendix A). In the case of T7, viral genes encode lysis proteins that act sequentially [17,28]. The first step is pore formation in the inner membrane, which leads to the release of peptidoglycan-digesting enzymes. At this stage, the membrane structure is expected to be similar to that proposed by Wong et al. [27]. The peptidoglycan structure becomes sporadically disrupted, which leads to membrane bulging at the defect points. The so-formed blebs become weak points where local membrane rupture is more likely to happen, which would finally lead to cell lysis (Appendix A). An open question regarding bleb formation is whether their appearance is a deterministic step in the infection cycle of T7 or a bacterial defensive response mechanism. Viral infection-induced blebbing has already been observed by other groups using fluorescent labeling techniques [18]. Upon the introduction of SyO, cell lysis is coupled with a huge fluorescent signal. This immediate increase in fluorescence is due to the instantaneous labeling of freshly released phage progeny. Fluorescence increase after bacterial lysis could also be the result of labeling the bacterial genome, however, this is unlikely as infection by T7 leads to host DNA degradation, which allows the efficient replication of the viral genome [29]. After about a minute, freshly released phages diffuse away from the dead host to seek further targets viable for infection.

## 4. Materials and Methods

### 4.1. Chemicals

Lysogeny broth (LB), glutaraldehyde grade I (GA) solution and poly-L-lysine (PLL) solution were from Sigma-Aldrich (St. Louis, MO, USA). Sytox Orange (SyO) was from Thermo Fisher Scientific (Waltham, MA, USA). Nitrogen 5.0 gas was from Linde Gáz Magyarország Zrt (Budapest, Hungary). Water was purified with a Milli-Q Integral 3 Water Production Unit (Merck Millipore, Billerica, MA, USA). Round mica sheets were from Ted Pella, Inc. (Redding, CA, USA). Purified T7 bacteriophages were a kind gift from the group of Gabriella Csík [30].

### 4.2. Bacterial Cell Growth and Purification

*E. coli* (ATCC 11303) bacterial cell cultures were grown in LB at 37 °C until OD_550_ of 1.0. OD_550_ value was determined by the measured absorbance of the undiluted bacterial culture at 550 nm wavelength with a Unicam UV4 spectrophotometer. Subsequently, the cells were collected by centrifugation. The pelleted cells were resuspended in PBS buffer, and centrifugation was repeated three times to exchange the growth medium for a more suitable buffer for proper immobilization of bacteria on poly-L-lysine (PLL)-coated mica and coverslip surface. 

### 4.3. Surface Preparation and Modifications

Coverslips were cleaned by 10 min sonication in acetone (50 W, 1 s ON, 1 s OFF), rinsed with purified water, and dried in a vacuum. 100 µL of PLL was dropped on either freshly cleaved mica or pre-cleaned coverslips (depending on the type of measurement), incubated for 20 min, washed with purified water, then dried in an N_2_ gas stream.

### 4.4. Bacterial Cell Immobilization

100 µL of *E. coli* cells diluted in PBS to OD_550_ of 0.1 were dropped on PLL-coated mica or coverslips, then the cells were allowed to adhere to the surface for 10 min at room temperature. Subsequently, the surface was repeatedly washed with LB to remove unbound bacteria. T7 bacteriophages were added to LB in a 50:1 ratio relative to the number of bacterial cells. Exchanging the buffer to LB and incubating at 37 °C was necessary for reinitiating bacterial growth while also enhancing viral host recognition. Upon changing the medium to PBS, bacteriophages still adhered to, and even lysed bacteria; however, the process was much slower and less reliable, thus all experiments which involved live observation of T7 host recognition were performed in LB.

### 4.5. Sample Fixation for High-Resolution AFM Imaging

The infection process was fixed at certain time points during the infection and imaged with AFM. Unbound bacteriophages were removed by first washing the infected, surface-immobilized cells with PBS. Bacterial immobilization and phage to host adhesion were stabilized, by 30 min incubation in 5% GA/PBS solution, which was followed once again by repeatedly washing the surface with PBS. Fixed samples were imaged immediately and were not stored following the AFM measurements.

### 4.6. TIRF Microscopy of Host Infection

Bacteriophage host recognition and DNA injection were observed by using a 100× oil immersion TIRF objective in an Olympus IX81 inverted microscope. Fluorescence images were captured with various frame rates using an Andor iXon EM + 885 EM-CCD camera. Imaging was performed with low-intensity phase-contrast illumination so that the unlabeled bacterial outline was apparent. 561 nm laser excitation was provided by an Omicron LightHUB DPSS laser. T7 bacteriophages were labeled one hour prior to experiments via their internal DNA with Sytox Orange (SyO) at a final concentration of 1 µM. This intercalator dye was chosen for its efficiency in labeling DNA without disturbing its translocation or inhibiting DNA-dependent enzyme functions [14,31]. TIRF image series were analyzed with Fiji and ImageJ software (ImageJ, NIH, Bethesda, MD, USA) [32]. Time stamps in the top right corner of the videos and image series show the time passed since the moment of initial phage addition (unless stated otherwise), which we refer to as the start of infection. Regarding the number of phage-to-bacterium binding statistics, n = 24 bacterial cells were analyzed.

### 4.7. Single-Phage DNA-Ejection Experiments

To investigate the behavior of individual bacteriophages, an in-house PDMS (poly (dimethylsiloxane)) microfluidic flow cell was designed and mounted on a coverslip. Phages were followed in a controlled flow environment by using TIRF microscopy. After flushing the flow cell with PBS, bacteriophages were injected and incubated to allow for adhesion to the coverslip surface. The sample was diluted to a final concentration at which only a few phages were visible in the field of view. Dilution was necessary to prevent the spatial overlap of ejected DNA molecules. Following phage adhesion, the remaining phages were flushed out with PBS. Imaging was performed in 100 nM SyO at typical flow rates of ~1 mm/s.

### 4.8. Two-Dimensionally Restricted Bead Movement Experiments

1 µL of fluorescent bead solution (Fluoresbrite carboxy microspheres NYO, d = 59 nm, Polysciences, Warrington, PA, USA) was pipetted on a coverslip, then a second coverslip was dropped on top of the bottom one. Capillary activity tightly pressed the two coverslips together so that a narrow chamber formed, wherein the movement of the beads was restricted to quasi-two dimensions.

### 4.9. Viral Target Search Tracking

Image sequences of TIRF microscopy experiments were analyzed by using the TrackMate ImageJ plugin (ImageJ, NIH, Bethesda, MD, USA) [33]. Particles of 0.5 µm apparent diameter were tracked throughout the recordings, and their X-Y positions were collected and analyzed.

### 4.10. AFM Imaging and Analysis

Topological features of infected bacteria were imaged in liquid at 25 °C by using an Asylum Research Cypher ES instrument (Oxford Instruments, Abingdon, UK). Sample surfaces were scanned in non-contact mode with Olympus BL-AC40TS cantilevers (8 nm average tip radius) oscillated by BlueDrive technology (photothermal excitation) near their resonance frequencies (~20 kHz) at typical scan speeds of 5 µm/s. Scanning resolutions were typically 512 pixels/line. Image post-processing and analysis were performed within the AFM driving software (IgorPro, WaveMetrics, Portland, OR, USA). Membrane roughness (Rq, root-mean-square roughness) was calculated on curvature-corrected 1 × 1 µm regions of bacterial surfaces, using the built-in function of the AFM driving software. Roughness was calculated by selecting multiple 0.5 × 0.5 µm regions of cell surfaces on at least three cells per fixation timepoints. Regarding the number of phage-to-bacterium binding statistics, n = 9 bacterial cells were analyzed.

### 4.11. Phase-Contrast Microscopy

Bacterial cell growth and lysis were observed at 100× magnification by using a Nikon Eclipse Ti inverted microscope. Image series were recorded at 5 s intervals using a Jenoptik Gryphax ProgRes camera.

## 5. Conclusions

Viral infection comprises a highly complex series of interconnected and synchronized events. T7 binds reversibly to the *E. coli* surface which it explores for the proper binding site by diffusive motion. Exploration proceeded with a constant speed; thus, our results suggest that this motion entails the rolling of the phage on the bacterial surface rather than walking with the fibers, which should be a decelerating process. Phage docking is initiated by binding of the tail to its receptor, and the fibers are involved in capsid positioning for rapid (~1 min) and efficient genome transfer. DNA release can be triggered in vitro by a photothermal mechanism which leads to partial genome ejection that stops short of the lysis-related genes, pointing at the presence of mechanical genome-transfer control. The T7 infection ends with violent cell lysis preceded by bacterial membrane surface roughening and bleb formation. Single virus observations such as the ones employed in this work can greatly contribute to the general understanding of nature’s nanomachines. As a future perspective, experiments employing structured illumination microscopy (SIM) combined with TIRF [34], as well as virus functionalized AFM tips combined with fast-force acquisition techniques might unravel important details of virus-host interactions [35].

## Figures and Tables

**Figure 1 ijms-23-11252-f001:**
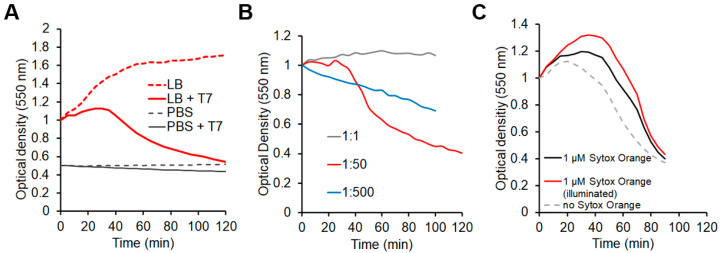
Bacterial growth kinetics: Optical density measurements corresponding to the density of the bacterial cultures over time. (**A**) Buffer composition influences bacterial growth. (**B**) Different MOIs have a significant impact on bacterial lysis. (**C**) Intercalator dyes delay and prolong infection-induced bacterial lysis. Optical density measurements were carried out in LB unless otherwise stated.

**Figure 2 ijms-23-11252-f002:**
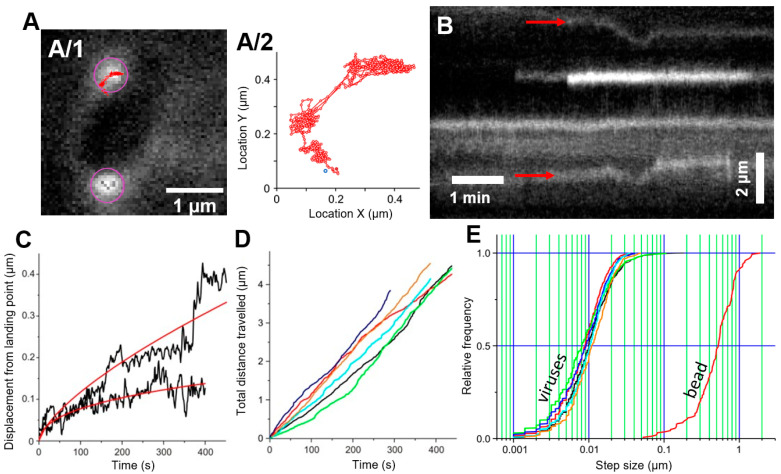
Phages travel on the bacterial surface. (**A**) Bacteriophages searching on the surface of a bacterium: (**A/1**) Purple circles show tracked particles, the red line shows the trajectory of the upper particle. (**A/2**) Enlarged graph of the trajectory shown in (**A/1**), the blue spot shows the initial binding point. (**B**) Kymogram of bacteriophages traveling along the bacterial periphery. Kymograms have been created from the elliptical periphery of individual bacteria (indicated by a white dashed line in Figure 3A). Red arrows show the initial docking position of two phages. (**C**) Black curves show the distance of two tracked viruses from their landing point over time, the red line shows a parabolic function fitted to the data with exponents of 0.64 and 0.43. (**D**) Total distance traveled by tracked phages from their initial binding point. Different colors correspond to different viruses. (**E**) Step size distribution of bacteriophages, and a 60 nm diameter bead. When comparing the step size distributions of viruses and that of the bead, it must be noted that the movement of phages was recorded at 1 frame per second, while the movement of beads was recorded at 10 frames per second.

**Figure 3 ijms-23-11252-f003:**
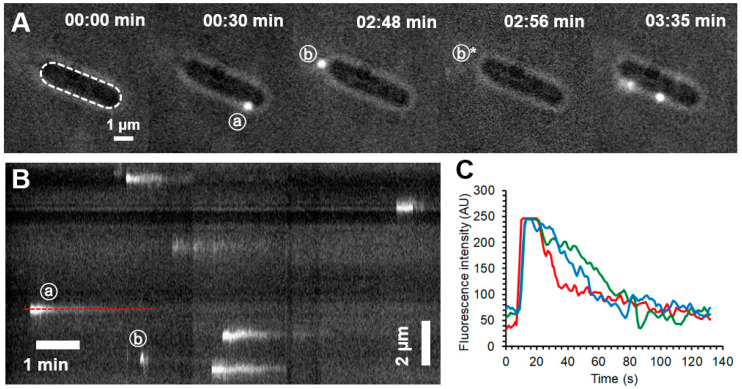
Single-phage infection process. (**A**) Snapshots of the early moments (0–10 min) of a single bacterium infection highlighting the docking of bacteriophages. The bacterial outline is slightly visible due to low levels of phase-contrast illumination. Dashed white line indicates the bacterial periphery along which a kymogram (**B**) was constructed. The same two phages are marked with circled (**a**) and (**b**) in the snapshots (**A**) and the kymogram (**B**). (**b***) marks the position where phage (**b**) was visible 8 s earlier. Red dashed line marks the location of the relative intensity profile of a single phage (**a**) shown in (**C**) in red. Blue and green lines show the injection kinetics of two additional phages.

**Figure 4 ijms-23-11252-f004:**
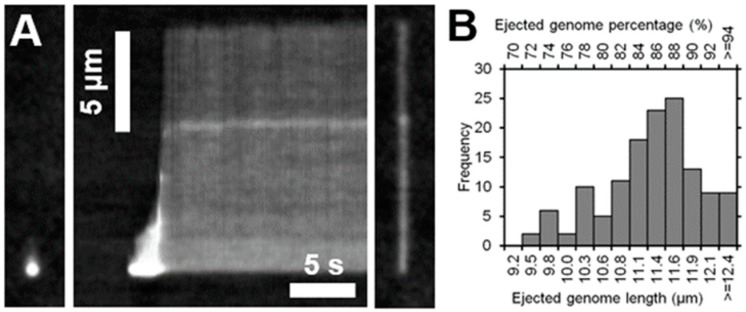
Bacteriophage DNA ejection in solution. (**A**) Images (left and right), and kymograph (middle) of a bacteriophage ejecting its DNA. (**B**) Distribution of ejected genome fractions.

**Figure 5 ijms-23-11252-f005:**
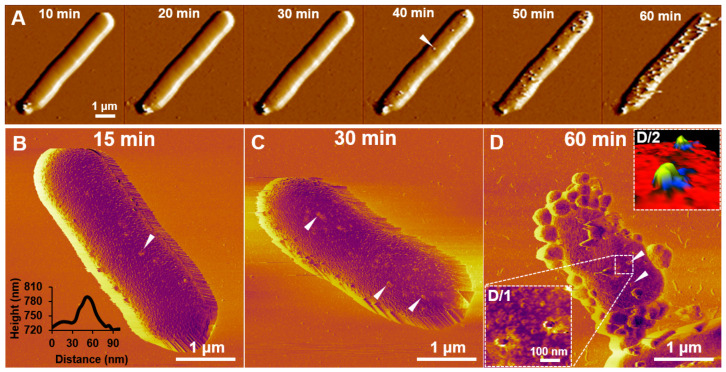
Viral infection causes bacterial cell surface roughening and bleb formation. (**A**) Amplitude-contrast image series of viruses docking on a single bacterial cell surface in native conditions at 25 °C. The white arrowhead points to a docked bacteriophage. Phase-contrast AFM images of *E. coli* bacteria fixed with glutaraldehyde 15 (**B**), 30 (**C**), and 60 (**D**) minutes post-infection. 60 min after infection bleb formation can be observed. White arrowheads point to docked bacteriophages. Insets show the enlarged region of two docked bacteriophages (**D/1**) and their 3D representations (**D/2**), yellow color shows the viral capsids and the black color shows the surface-bound tail fibers.

**Figure 6 ijms-23-11252-f006:**
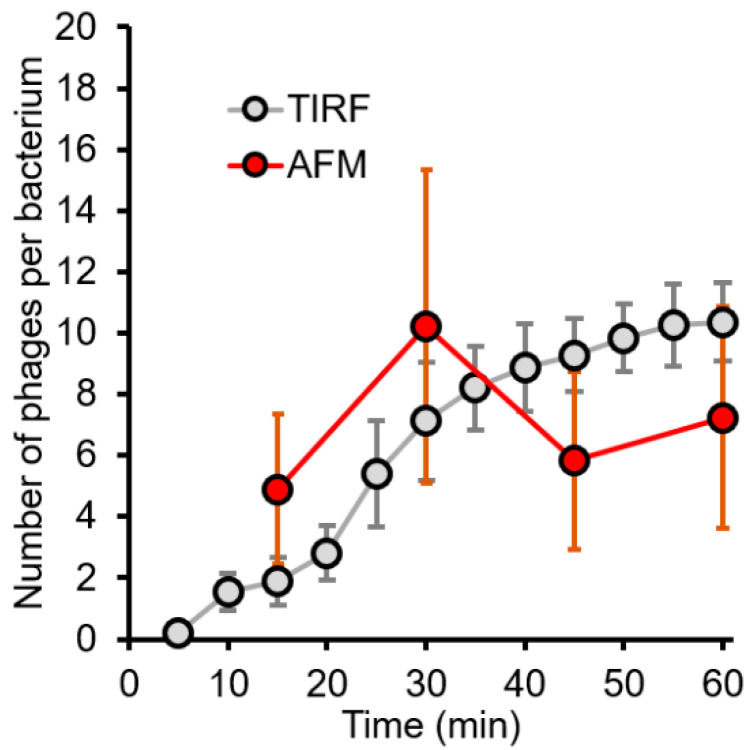
Bacteriophages saturate the bacterial surface over time. The number of docked bacteriophages followed by TIRF and AFM. n = 24 for TIRF and 9 for AFM.

**Figure 7 ijms-23-11252-f007:**
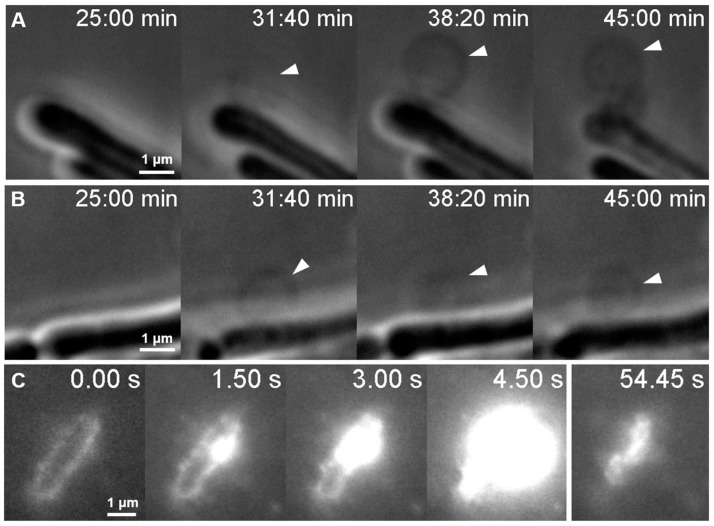
Viral infection induces blebbing and leads to explosive lysis. (**A**) and (**B**) show two exemplary phase-contrast image series of bacterial blebbing. (**C**) A TIRF microscopy image series depicting certain moments from the lysis of a single bacterium in the presence of SyO. Timestamp starts from 35 min post-infection.

## Data Availability

Not applicable.

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
