# Peer review of "Imaging the Infection Cycle of T7 at the Single Virion Level"

_ijms, 2022, doi:10.3390/ijms231911252_

Round 1

Reviewer 1 Report

The article from Kiss et al. presents a comprehensive study to understand the key steps of the T7 infection cycle on E. coli. The authors used a range of single-particle microscopy methods like TIRF, AFM and phase contrast microscopy to understand each step of the T7 virus-host infection steps and highlighted some interesting observations on some of the associated dynamic processes. Using TIRF microscopy, they investigated the dynamics of phase docking and DNA ejection into the host E. coli. Using AFM, they looked at the structural dynamics of fixed host cell samples at different time periods of phase infection and lysis. Further, they used phase contrast microscopy to the dynamics of bleb formation on infected E. coli cells resulting lysis. Overall, their work highlights the visualization and quantification some of the key steps of the phase infected E. coli.

From the standpoint of visualizing these key steps, the study is extensive, well written and provides a necessary framework for looking into viral infection steps. However, the authors are unclear on providing a conclusive and comprehensive model for the infection steps using key findings from the microscopy observations. For example, in one such case for phase dynamics on bacterium surface, the authors conclude deceleration of phases over time as well as phase motion with constant speed. While the results seem more in line with later, the conclusions are not quite clear. Because of these factors, the paper cannot be accepted in its current form, and needs further revision and conclusive explanations of results to warrant publication.

Here are some comments to be considered for improving the manuscript.

1.   Models for each observations need more analysis and conclusions. For. e.g (page 6, line 230-236) from 2D velocity of phase motion on bacterium looks more or less constant (for different colors?). However, the observation of phase exploration process being progressively decelerating is not very clear.

2.      Figures’ description in a few cases are incomplete and did not provide necessary information to understand the figures. For e.g. In Figure 1 and 2 (page 6, line 225), the description is incomplete, making it harder to fully understand the context (What does different colors in 2D and 2E correspond to?). Additionally, figures are not in line with the texts and have repeated figure descriptions.

3.      Looking at the fate of docked T7 (page 6, line 242) the authors characterized 3 different processes (DNA injection into target, DNA misfiring and phase dissociation). While the author mentioned the first event is most common, what are the probabilities of occurrence of these events and on what factors do they depend on?

4.      What does the multi- exponential nature of fluorescence intensity decay mean (page 6, line 248)? Have the authors done any quantification of the exponential decay of fluorescence (figure 3C)?

5.      The events in the kymographs needs more clear explanation (figure 2 & 3). How do the authors differentiate phase diffusion/motion vs dissociation?

6.      Looking at the phase kinetics (association and dissociation rates) on the cell surface from the resulting observations and how they may vary with different conditions, for e.g. different phases (and their tails) would be crucial in characterizing these processes.

7.      The authors describe not seeing an increase in fluorescence intensity upon DNA injection is due to the escape of DNA in the FOV of TIRF or dislocation of sytox-O by DNA-dependent polymerases (page 7, line 263). Are there any control studies (microscopic or mutational) to validate this observation? Wide field/epifluorescence microscopy may confirm the first hypothesis, whereas mutating the polymerases may validate the second.

8.      Are there other studies or evidence to support that AFM as a mechanical disinfectant for virus-host infections (Page 9, line 324)? Can such knocking off of virus particles be due to other factors, like fixing methods or measurement conditions, impacting strength of virus-host interactions?

9.      E. coli in Italics (page 9, line 331)

10.  What is the method for calculating roughness, Rq values?

11.  In figure 6, decrease in the number of phases per bacterium for AFM measurements, can be due to AFM as a molecular disinfectant and/or non-uniform topological detection due to bleb. For early stage of fixing (say 15 mins) virus blebs seem not to have formed on infected bacterium. However, due to the nature of AFM measurements for removing docked virus particles, number of virus particles per bacterium should be less then what observed in TIRF. However, the numbers for AFM in 15 and 13 mins seems higher them TIRF, which is in contradiction. What may be the reason for such observation?

12.  In conclusion (page 12, line 409-410), how are the authors differentiate rolling vs walking motion of the phase on the bacteria surface? Which microscopic events particularly point towards such behaviors?

13.  There are two figure titles for each figure, except for figure 2.

14.  The text format for the main article is different from the references.

Reviewer 2 Report

The authors report an experimental study of T7 phages infection where they take advantage of TIRF and AFM technique.

The paper describe in great detail the variuos experiments performed by the author, with great experimental skill.

The results are well presented and the discussion is clear and convincing.

Self citation are certainly inherent to the work and limited in number.

There are layout problems with respect to the captions of almost all the figures. The caption text below the figure is cut off and the same text is repeated immediately below. Marked in the attached pdf.

In conclusion I consider this a good quality work and recommend it for publication.

Reviewer 3 Report

The manuscript titled “Imaging the infection cycle of T7 at the single virion level” by Kiss, B.; et al. is an original work where the authors address the viral infection of E. Coli covering its subsequent stages (bacteriophage docking, translocation of DNA and final bacterial lysis). Authors succeed to combine multidisciplinary techniques like turbidimetry, total internal reflection fluorescence (TIRF) microscopy and atomic force microscopy (AFM). These latest two single molecule techniques enable to decipher the hinder phenomena associated to bulk measurements taking into account several environmental conditions as phage-to-cell-ratio, culture media nature or the DNA-staining dye concentration. Finally, authors propose a molecular mechanism of T7 viral infection on E.Coli that could be extrapolated to other viral pathogen-host systems. The gathered findings may be relevant for the examined field. The results achieved are well-discussed during the main body of the reported manuscript. The scientific paper is well written. In my opinion the present manuscript is innovative and the methodological approached used matches with the scope of International Journal of Molecular Sciences. For the above described reasons, I recommend the publication in International Journal of Molecular Sciences once the following remarks will be fixed:

--------

KEYWORDS

The selected keywords are the appropriated respecting the content of the submitted work. Nevertheless, I may consider to merge “total internal reflection fluorescence” and “atomic force microscopy” terms with TIRF and AFM, respectively. Then, TIRF and AFM should appear between brackets.

--------

INTRODUCTION

Introduction section is clear and concise. It lacks some further details about potential applications found related of the most interesting findings. Authors briefly tackle this point during the Discussion section: “Single virus observations such as the ones employed in this work can greatly contribute to the general understanding of nature’s nanomachines” (lines 416-418).

Some statements should be added in the Introduction section on this regard related to viral nanomachines [1], design treatments against antibiotic resistance diseases [2] or other many biotechnological applications [3] like agriculture and food industries.

[1] Hemminga, M.A.; Vos, W.L.; Nazarov, P.V.; Koehorst, R.B.M.; Wolfs, C.J.A.M.; Spruijt, R.B.; Stopar, D. Viruses: indredible nanomachines. New advances with filamentous phages. Eur. Biophys. J. 2010, 39, 541-550. https://doi.org/10.1007/s00249-009-0523-0.

[2] Nick, J.A.; Dedrick, R.M.; Gray, A.L.; Vladar, E.K.; Smith, B.E.; Freeman, K.G.; Malcolm, K.C.; Epperson, L.E.; Hasan, N.A.; Hendrix, J.; et al. Host and pathogen response to bacteriophage engineered against Mycobacterium abscessus lung infection. Cell 2022, 185, 1860-1874. https://doi.org/10.1016/j.cell.2022.04.024.

[3] Varanda, C.; Félix, M.; Campos, M.D.; Patanita, M.; Materatski, P. Plant Viruses: From Targets to Tools for CRISPR. Viruses 2021, 13, 141. https://doi.org/10.3390/v13010141.

--------

MATERIALS & METHODS

Some further information must be provided in this section.

Bacterial cell growth and purification (line 89). Authors should specify the UV-Vis equipment used to obtain the absorbance from the tested bacterial cultures.

Sample fixation for high-resolution AFM imaging (line 111). Did the authors prepare the decorated surfaces with E.coli prior the AFM measurements? Or these surfaces were stored at certain conditions (e.g. 4 ºC, overnight) upon AFM measurements?

TIRF microscopy of host interaction (line 117). “TIRF image series were analyzed with Fiji and ImageJ software” (lines 126-127). Authors should add the following reference [4] for Image J software:

[4] Schneider, C.A.; Rasband, W.S.; Eliceiri, K.W. NIH Image to ImageJ: 25 years of image analysis. Nat. Methods 2012, 9, 671-675. https://doi.org/10.1038/nmeth.2089.

Single-phage DNA ejection experiments (line 130). Please, define the term “PDMS” (line 131) as poly(dimethylsiloxane).

Two-dimensionally restricted bead movement experiments (line 139). Please, authors should indicate the supplier details of “Fluoresbrite carboxy microspheres” (line 140).

AFM imaging and analysis. Authors should indicate the nominal AFM tip radius (8 nm) and the image resolution during acquisition (pixels/line). Moreover, the number of AFM measurements (population size, N) to calculate the bacterial membrane roughness must be also provided.

--------

RESULTS

The most significant outcomes are perfectly explained for all potential target audiences and other stakeholders. Nevertheless, the following points should be addressed:

I)          T7 explores the E.coli surface for binding site. “To follow the target recognition kinetics (…) their DNA was labelled with SyO (…)” (lines 184-185). What is the lower detection limit of SyO in terms of base pairs (bp)? Authors should state this information.

II)       Authors compare the diffusion process between viral particles and polysterene beads as negative control (lines 199-213). Motion differences were observed when these two different systems are compared. Authors hypothesize that the greater speed found for polysterene beads proofs the lack of transient interactions between the polysterene bead and the bacteria surfaces. I’m agree with this observation but could also the differences in terms of density between viral and polysterene particles play a role on the gathered results?

III)    Figure 3 (line 255). The Figure is nice and informative. It may convenient to introduce an extra section letter (D) carrying out statistics with several bacteriophages trajectories in the same plot (e.g. N = 10). This implementation will significantly aid to potential readers to understand if the trajectory related with the fluorescence decay is consistent.

IV)    Figure 5 (line 329). “(…) E. coli bacteria fixed with glutaraldehyde”. Could fixation procedure negative affect bacteria membrane integrity? Authors should provide some further information on this regard. Moreover, the Figure 5 caption is duplicated. The upper caption should be erased.

V)       Figure 5B. Authors mix qualitative information from the phase (entire image) with quantitative information provided by the crossection profile (white triangle). I can understand the preference to use phase images in Fig. 5A to gain clarity but, I strongly encourage the authors to depict topography images (displaying their respective scale bars) in Fig. 5 B-D.

VI)    Figure 6 (line 366). Same problem as aforementioned. It appears two figure captions. One of them should be erased. Moreover, the population size (N = 24) should be also indicated in the respective Material & Methods section. Same for Figure 7 (line 393).

--------

CONCLUSION

This section encompasses the main findings of the present manuscript. It only lacks recent developed single molecule methodologies and approaches that could be used as future perspectives in this field. In this framework, TIRF microscopy can be combined with structured illumination microscopy (SIM) emerging TIRF-SIM that enables the simultaneously acquisition of molecular dynamics and fluorescent lifetime kinetics under relevant microenvironmental conditions [5]. Accurate analysis of local AFM images setting defined height threshold masks that makes negligible any interference by background subtraction [6]. This fact enables precise characterization of biomolecular feature morphologies (e.g. bacteria membranes or virus capsids). Finally, the design and optimization of fast-force curve acquisition with functionalized AFM tips [7]  that can lead future quantitative molecular recognition imaging of bacterial membrane receptors.

[5] Richter, V.; et al. Super-Resolution Live Cell Microscopy of Membrane-Proximal Fluorophores. Int. J. Mol. Sci. 2020, 21, 7099. https://doi.org/10.3390/ijms21197099.

[6] Marcuello, C.; Frempong, G.A.; Balsera, M.; Medina, M.; Lostao, A. Atomic Force Microscopy to Elicit Conformational Transitions of Ferredoxin-Dependent Flavin Thioredoxin Reductases. Antioxidants 2021, 10, 1437. https://doi.org/10.3390/antiox10091437.

[7] Marcuello, C.; de Miguel, R.; Lostao, A. Molecular Recognition of Proteins through Quantitative Force Maps at Single Molecule Level. Biomolecules 2022, 12, 594. https://doi.org/10.3390/biom12040594.

--------

REFERENCES

Bibliography citations are not in the proper format of International Journal of Molecular Sciences. Authors should take care of this aspect.

--------

OVERVIEW AND FINAL COMMENTS

The submitted work is well-designed and the gathered results are interesting for the design and fabrication of next-generation of therapies against bacterial diseases. The gathered know-how of underlying molecular mechanisms of bacterial infection by bacteriophages can shed light on this topic. For this reason, I will recommend the present scientific manuscript for further publication in International Journal of Molecular Sciences once all the aforementioned suggestions will be properly fixed.

Round 2

Reviewer 1 Report

The revised manuscript from Kiss et al, has carefully addressed my questions. There are a few minor typos, that can be addressed in the final version of the manuscript. I do not have any further comments and recommend its publication after some minor revisions.

1-     Page 3, line 123 -  “measruements” is spelled incorrectly.

2-     Figure 5, description  - “P hase” in Phase contrast is a single word; “reg ion” is a single work.

3-     Figure 6, description -  No dash between “over-time”

4-     Page 12, line 402 -  “actions” would be singular

5-     Why some references have DOIs (#12,14,15), whereas others don’t (from #1-7)? All references should be in the same format as per MDPI guidelines.

Author Response

We would like to once again thank the reviewer for the thorough reading of our paper!

Our responses to the raised questions are the following:

  1. The misspelling is now corrected.
  2. These two typos are actually formatting errors which were only present in the pdf version of the document. The width of the caption box has now been changed to correct for these errors. Hopefully the final version won't have this issue now.
  3. Thank you for noticing this, now the description states "over time", without the dash.
  4. "Action" is now in its singular form, thank you for noticing this!
  5. The references have now been all formatted in the same style, the corresponding DOIs have been added to all the references where it was missing.

Reviewer 3 Report

The authors have fully fulfilled my requirements. For this reason, I warmly recommend this work for further publication in the International Journal of Molecular of Molecular Sciences (IJMS) once the reference citations are in the proper IJMS format.

Author Response

We would like to thank the reviewer for the recommendation of our work for publication, the references now appear in the style provided by MDPI, furthermore the missing DOI numbers have now been added to the relevant references.